# USO: Unified Style and Subject-Driven Generation via Disentangled and Reward Learning

## Abstract

Existing literature typically treats style-driven and subject-driven generation as two disjoint tasks: the former prioritizes stylistic similarity, whereas the latter focuses on subject consistency, resulting in an apparent antagonism. We argue that both objectives can be simultaneously achieved within a unified framework, as they fundamentally pertain to the disentanglement and re-composition of *content* and *style*, a longstanding theme in both tasks. To this end, we introduce **USO**, a **U**nified **S**tyle-subject **O**ptimized customization model that leverages the complementary nature of these objectives, enabling them to mutually reinforce and enhance each other within a cohesive paradigm. Specifically, on the one hand, we first propose a *subject-for-style* data curation framework that leverages a state-of-the-art subject model to generate high-quality triplet data comprising content images, style images, and their corresponding stylized content images. Building on this foundation, USO further introduces a *style-for-subject* approach for content-style disentangled learning, which simultaneously aligns style features and content features to construct a cohesive customization model. Furthermore, a style reward-learning, termed SRL, is further applied to reinforce the model's ability to extract desired style or content features from the reference image, thereby further enhancing the performance of both tasks. Extensive experiments demonstrate that USO achieves state-of-the-art performance among open-source models along both dimensions of subject consistency and style similarity.

## 1 Introduction

The significant advancements in image generation over the past years have greatly improved generative controllability, fundamentally changing how humans create images, *i.e.*, whether through abstract textual descriptions, specific visual reference images, or both. Research on leveraging both textual and visual conditions has attracted increasing interest, giving rise to numerous real-world tasks such as style-driven generation and subject-driven generation. While textual conditions are typically explicit, ***visual conditions are inherently noisy***, as images intrinsically embody a rich spectrum of features (*e.g.*, style, appearance), of which only a specific one is relevant to a specific task. For instance, style-driven generation requires only the style feature from the reference images, whereas other features constitute noise. Therefore, a fundamental and long-standing challenge in these tasks is to accurately ***include all required features from the reference image while simultaneously excluding other noisy ones***, *e.g.*, including only the style in style-driven generation or only the subject's appearance in subject-driven generation.

Extensive efforts in the literature have been dedicated to disentangling different features in visual conditions (*i.e.*, reference images). On the one hand, in the realm of style-driven generation, DEADiff Qi et al. (2024) employs QFormer to selectively query only the style features from reference images. CSGO Xing et al. (2024) constructs content-style-stylized triplets to facilitate style-content decoupling during training. StyleStudio Lei et al. (2025) introduces style-based classifier-free guidance (SCFG) to enable selective control over stylistic elements and to mitigate the influence of irrelevant features. On the other hand, subject-driven generation methods primarily focus on disentangling subject appearance features or constructing more effectively disentangled paired data. For example, RealCustom Huang et al. (2024b); Mao et al. (2024) proposes a dual-inference framework that selectively incorporates subject-relevant features into subject-specific regions. UNO Wu et al. (2025c) leverages the in-context capabilities of DiT to progressively improve both the quality of

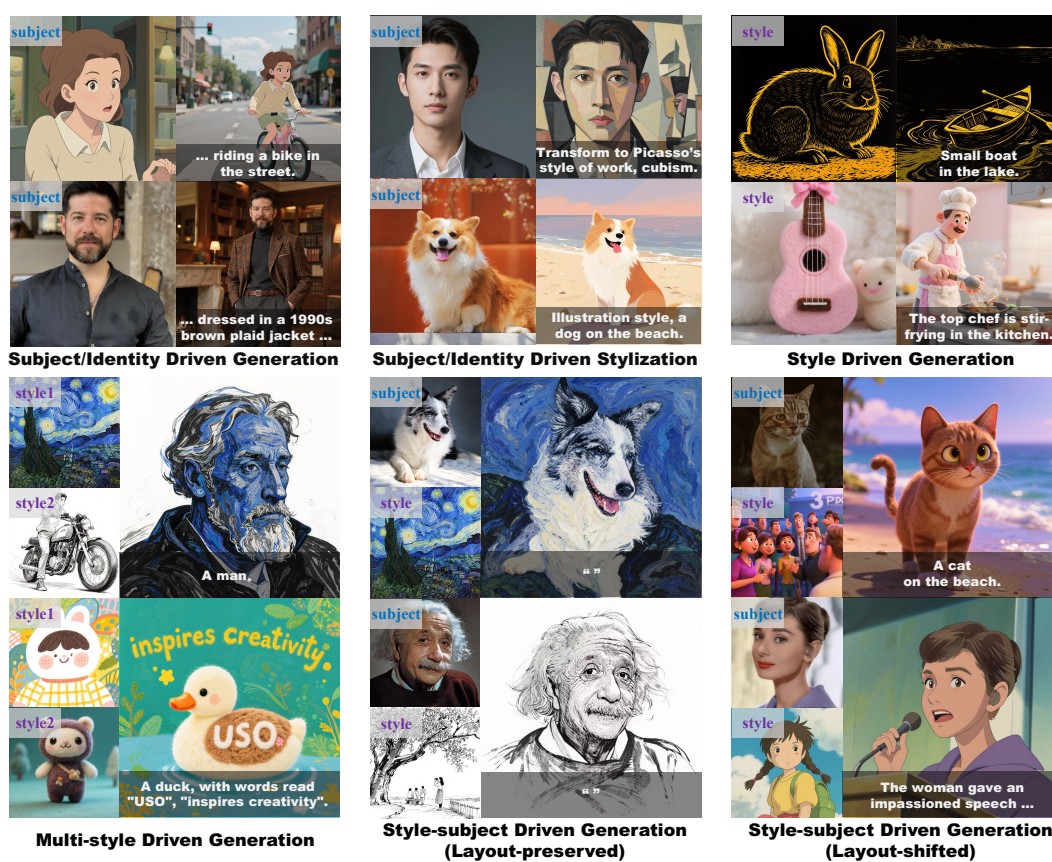

Figure 1: We propose **USO**, a unified model that jointly optimizes for subject and style, enabling customizable generation with high subject consistency and style fidelity.

paired data and the model itself. To conclude, existing methods primarily focus on ***task-specific disentanglement*** by designing tailored datasets or model architectures for each individual task, thereby performing ***disentanglement in an isolated, single-task context.***

In this study, we argue that a more comprehensive and precise disentanglement approach should fully account for the coupling and complementarity between different generation tasks. Each task should not only learn which features to include, but, more importantly, also learn which features to exclude, *i.e.*, features that are often required by other tasks. ***Therefore, learning to include certain features in one task inherently informs and enhances the process of learning to exclude those same features in a complementary task, and vice versa.*** For example, style-driven generation aims to incorporate stylistic features while excluding subject appearance features, whereas subject-driven generation does the exact opposite. The ability to learn and include subject appearance features in subject-driven generation can, in turn, help style-driven generation more effectively learn to exclude these features, thereby improving disentanglement for both tasks. In conclusion, we believe that jointly modeling complementary tasks enables a mutually reinforcing disentanglement process, leading to a more precise separation of relevant and irrelevant features for each task.

Based on the above motivation, we propose a novel ***cross-task co-disentanglement*** paradigm to unify subject-driven and style-driven generation, and, more importantly, to mutually enhance the performance of both tasks. Specifically, this co-disentanglement paradigm is implemented through a *subject-for-style* data curation framework and a *style-for-subject* model training framework. The *subject-for-style* framework first utilizes a state-of-the-art subject model to generate high-quality style data, while the *style-for-subject* framework subsequently trains a more effective subject model under the guidance of style rewards and disentangled training. Technically, on the one hand, for the *subject-for-style* data curation framework, we build upon a state-of-the-art subject-driven model Wu et al. (2025c) and further develop both a stylization expert and a de-stylization expert to curate stylized and non-stylized images. This process ultimately constructs triplet data pairs in the form of <style reference, de-stylized subject reference, stylized subject result> for subsequent model train-

ing. On the other hand, for the *style-for-subject* model training framework, we propose a **U**nified **S**tyle-**S**ubject **O**ptimized (**USO**) customization model, which introduces style-subject disentanglement training and style reward learning.

Our contributions are summarized as follows:

**Concept:** We point out that existing style-driven and subject-driven methods focus solely on isolated disentanglement within each task, neglecting their potential complementarity and thus leading to suboptimal disentanglement. For the first time, we propose a novel cross-task co-disentanglement paradigm that unifies style-driven and subject-driven tasks, enabling mutual enhancement and achieving significant performance improvements for both.

**Methodology:** We present a novel cross-task triplet curation framework that bridges style-driven and subject-driven generation. Building on this, we introduce USO, a unified customization architecture that incorporates content–style disentanglement training and a style reward learning paradigm to further promote cross-task disentanglement. We further release USO-Bench, to the best of our knowledge, the first benchmark tailored for evaluating cross-task customization.

**Performance:** Extensive evaluations on USO-Bench and DreamBench Ruiz et al. (2023) show that USO achieves state-of-the-art results on subject-driven, style-driven, and joint style-subject-driven tasks, attaining the highest CLIP-T, DINO, and CSD scores. USO can handle individual tasks and their free-form combinations while exhibiting superior subject consistency, style fidelity, and text controllability as shown in Figure 1.

## 2 RELATED WORK

### 2.1 STYLE TRANSFER

Style Transfer aims to apply the style in the reference image to the given content image or fully generated image. Early work like adaptive instance normalization Huang & Belongie (2017) achieved impressive style transfer results with layout-preserved results by simply using a pre-trained network as the style encoder and well-designed injection modules. The recent powerful text-to-image generation base models, like Stable Diffusion Podell et al. (2024); Esser et al. (2024) and FLUX Labs (2024), along with style transfer plugins built upon them, have significantly improved the convenience and effectiveness of performing this task. Several are even training-free, like StyleAlign Wu et al. (2021) and StylePrompt Jeong et al. (2024) which transfer the style via simple query-key swapping in the specific self-attention layers. Other training-based methods can theoretically achieve better fitting and style transfer performance, but they also raise concerns of content leakage. IP-Adapter Ye et al. (2023) and DEADiff Qi et al. (2024) demonstrate the style transfer ability with a new decoupled cross-attention layer trained with coupled data, and overcome the content leakage by decreasing the injection weights in inference-time. InstanceStyle Wang et al. (2024), StyleShot Gao et al. (2024) and B-lora Frenkel et al. (2024) provide more detailed time-aware and layer-aware injection strategies to disentangle the style and content feature injections. However, those disentanglement analyses are tied to the specific model architecture and hard to migrate.

### 2.2 SUBJECT-DRIVEN GENERATION

Subject-driven generation refers to generating images of the same subject conditioned on a text instruction and reference images of given subjects. Dreambooth Ruiz et al. (2023) and IP-Adapter Ye et al. (2023) turn a UNet-based text-to-image model into a subject-driven model by parameter-efficient tuning or a newly introduced attention plug-in. Recently, popular image-generation foundation models have shifted from UNet-based architectures to transformer-based ones. The inherent in-context learning capabilities of transformers have greatly enriched research on subject-driven generation. ICLoRA Huang et al. (2024a), OminiControl Tan et al. (2024), UNO Wu et al. (2025c), and FLUX.1 Kontext Labs et al. (2025) use shared attention between the generated image and reference image to train a text-to-image DiT into a subject-driven variant. It is worth noting that some of them have extended the reference subject to other types. OminiControl Tan et al. (2024) supports layout control image as a reference, UNO Wu et al. (2025c) supports multiple reference images input, and DreamO Mou et al. (2025) can work for simple style transfer. They have indicated that various types of reference-guided generation can be unified within the DiT in-context framework.

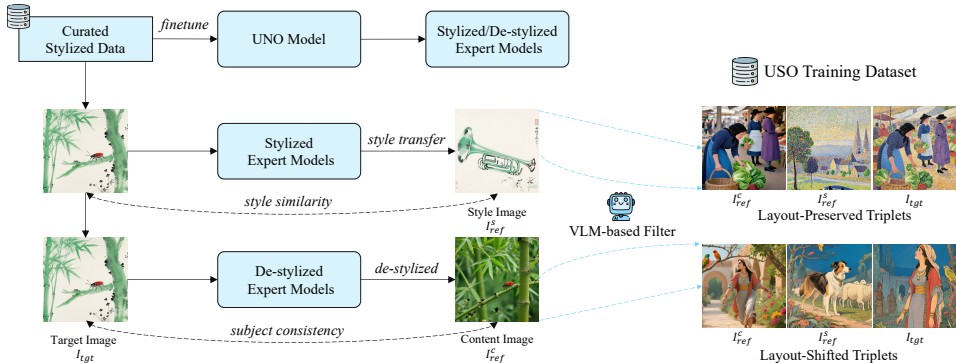

Figure 2: Illustration of our proposed cross-task triplet curation framework, which systematically generates layout-preserved and layout-shifted triplets.

This further prompts the question of whether jointly addressing different tasks in this setting could lead to mutual benefits across them.

## 3 METHODOLOGY

### 3.1 CROSS-TASK TRIPLET CURATION FRAMEWORK

This section details the construction of cross-task triplets for USO training. Although prior works Xing et al. (2024); Wang et al. (2025) have explored triplet generation, they retain the original layout, preventing any pose or spatial re-arrangement of the subject. To jointly enable subject-driven and style-driven generation beyond simple instruction-based edits, we curate a new USO dataset expressly designed for this unified objective.

Figure 2 provides an overview of USO dataset. Our co-disentanglement paradigm starts from a *subject-for-style* data curation framework. Among many possible tasks, subject-driven (i.e., UNO-1M Wu et al. (2025c)) and instruction-based editing (i.e., X2I2 Wu et al. (2025b)) datasets are comparatively easy to collect at scale, enabling targeted task-specific corpora. In particular, subject-driven data emphasizes learning from content cues while preserving subject identity and consistency; instruction-based editing bridges styles by preserving spatial layout and transferring appearance between realistic and stylized domains in both directions. These resources naturally support training domain-specialist models and, through deliberate dataset design, induce the capabilities we care about (*e.g.*, extracting task-relevant features conditioned on image type). Guided by these insights, we curate $200k$ stylized image pairs sourced from publicly licensed datasets and augmented with samples synthesized by state-of-the-art text-to-image models. Using these data, we trained two complementary experts on top of the leading customization framework UNO Wu et al. (2025c): **(1) a stylized expert model** that performs style-driven generation conditioned on a style-reference image, producing a new subject rendered in the target style ($I_{\text{ref}}^s$ from $I_{\text{tgt}}$), and **(2) a de-stylization expert model** that inverts a stylized image to a photorealistic counterpart, allowing either flexible layout shifts or preservation ($I_{\text{ref}}^c$ from $I_{\text{tgt}}$).

Each curated stylized image serves as the target $I_{\text{tgt}}$. We synthesize its style reference $I_{\text{ref}}^s$ via the stylization expert and its content reference $I_{\text{ref}}^c$ via the de-stylization expert. Following Wu et al. (2025c), a VLM-based filter enforces style similarity between $I_{\text{tgt}}$ and $I_{\text{ref}}^s$ and subject consistency between $I_{\text{tgt}}$ and $I_{\text{ref}}^c$. This yields two kinds of triplets, shown in Figure 2: layout-preserved and layout-shifted. Unlike prior work Xing et al. (2024); Wang et al. (2025), which focuses solely on style-driven generation and confines itself to layout-preserved triplets, our cross-task triplets achieve deeper content–style disentanglement across tasks and are used to train USO.

### 3.2 UNIFIED CUSTOMIZATION FRAMEWORK (USO)

In this section, we describe how we unify two tasks that have traditionally been treated separately, subject-driven and style-driven generation, into a single model. we train USO on two kinds of triplets from Section 3.1. Critically, in addition to layout-preserved triplets, we introduce layout-shifted triplets—pairs where the spatial configuration changes, which are essential for a robust uni-

fied model: they force the network to inject the desired stylistic features while keeping the subject consistent across diverse stylized scenarios and varied text prompts.

### 3.2.1 CONTENT-STYLE DISENTANGLEMENT TRAINING.

**Disentangled conditional encoder.** As illustrated in Figure 3, We start from a pre-trained text-to-image (T2I) model and fine-tune it into a text-image-to-image (TI2I) model.

Unlike prior in-context generation approaches that rely solely on a VAE $\mathcal{E}(\cdot)$ to encode the conditioned image $I_{\text{ref}}$, we argue that style is a more abstract cue demanding richer semantic information. Therefore, we employ the semantic encoder SigLIP instead of the VAE to process the reference style image $I_{\text{ref}}^s$. While subject-driven or identity-preserving tasks typically emphasize high-level semantics, style-driven tasks must simultaneously handle two extremes: high-level semantics to accommodate large geometric deformations (e.g., 3-D cartoon styles) and low-level details to reproduce subtle brushstrokes (e.g., pencil sketches). Following recent works like Zhang et al. (2024), we introduce a lightweight Hierarchical Projector $\mathcal{M}_{\text{Proj}}(\cdot)$ to project multi-scale, fine-grained visual features $z_s$ from the extracted SigLIP embeddings $\{c_i\}_{i=1}^N$, where $N$ represents the layer indices of SigLIP. This process can be formulated as:

Figure 3: Illustration of the training framework of **USO**.

$$z_s = \text{Concatenate}(\mathcal{M}_{\text{Proj}}(\{c_i\}_{i=1}^N)), \qquad (1)$$

Then we introduce subject conditioning as shown in Figure 3. Following recent paradigms Tan et al. (2024); Wu et al. (2025c), the content image $I_{\text{ref}}^c$ is encoded into pure conditional tokens $z_c$ by a frozen VAE encoder $\mathcal{E}(\cdot)$. We formulate USO as a multi-image conditioned model, yet explicitly disentangle content and style features via separate encoders.

**Stochastic conditioning dropout training.** During training, we unfreeze the Hierarchical Projector and fine-tune the DiT with LoRA as shown in Figure 3. With probability $p$ we randomly drop either the style or the subject reference, forcing the model to solve pure subject-driven generation or pure style transfer tasks. This strategy preserves single-task capability while simultaneously exposing the network to a multi-task regime, enabling end-to-end learning of disentangled representations. The final multimodal input sequence $z_2$ is therefore expressed as:

$$z_2 = \text{Concatenate}(z_s, c, z_t, z_c), \qquad (2)$$

We set $p = 0.25$ during training. Style tokens $z_s$ are assigned the same positional indices as the text tokens $c$, while content tokens obtain their positions via UnoPE Wu et al. (2025c) using its diagonal layout. Consequently, USO can seamlessly handle both subject-driven and style-driven tasks on the proposed triplet dataset.

### 3.2.2 STYLE REWARD LEARNING

Although the above pipeline already formulates a unified customization model, one of our key insights is that learning to include desired features for one task helps the complementary task suppress those undesired features, thereby improving overall performance. To this end, we introduce Style Reward Learning (SRL) to boost style similarity and observe how it contributes to subject consistency. SRL alternates between computing a reward score and back-propagating the reward signal. Unlike traditional ReFL Xu et al. (2023), which in text-to-image generation primarily considers text fidelity or visual appeal, SRL is tailored for the reference-to-image setting. It focuses on reinforcing the model to extract desired features from a reference image by directly computing a reward between the online outputs and the conditioning image. As shown in Figure 3, we define the reward score as the style similarity between the reference style image $I_{\text{ref}}^s$ and the generated stylized image $\hat{I}_0$,

---

**Algorithm 1** Style Reward Learning (SRL) with Flow Matching

---

**Require:** Customization model `net` with pretrained parameters $\theta$; pretrain loss $\mathcal{L}_{\text{Pre}}$; reward loss $\mathcal{L}_{\text{SRL}}$; reward model $\mathcal{M}_{\text{RM}}$; balancing coefficient $\lambda$; noise-schedule steps $T$; SRL fine-tuning interval $[t_s, t_e]$; dataset $\mathcal{D} = \{(y, I_0, I_{\text{ref}}^c, I_{\text{ref}}^s)\}$, $y$ is prompt, $I_0$ is target image and $I_{ref}^c, I_{ref}^s$ are reference content and style images (Section 3.1)

1: **for** $(y, I_0, I_{\text{ref}}^c, I_{\text{ref}}^s) \in \mathcal{D}$ **do**
2:  $\quad \mathcal{L}_{\text{Pre}} \leftarrow \text{net}_\theta(y, I_0, I_{\text{ref}}^c, I_{\text{ref}}^s)$ // calculate pretrain loss with Equation (4)
3:  $\quad t \sim \mathcal{U}(t_s, t_e)$ // pick a random time step in $[t_s, t_e]$
4:  $\quad x_T \sim \mathcal{N}(\mathbf{0}, \mathbf{I})$
5:  $\quad$ **for** $\tau = T, \ldots, t+1$ **do**
6:  $\quad\quad \hat{v}_\tau \leftarrow \text{no-grad}(\text{net}_\theta(y, x_\tau, I_{\text{ref}}^c, I_{\text{ref}}^s))$
7:  $\quad\quad x_{\tau-1} \leftarrow x_\tau - \hat{v}_\tau \Delta t$ // reverse-step update
8:  $\quad$ **end for**
9:  $\quad \hat{v}_t \leftarrow \text{net}_\theta(y, x_t, I_{\text{ref}}^c, I_{\text{ref}}^s)$
10: $\quad \hat{I}_0 \leftarrow \text{decode}(x_t - \hat{v}_t \Delta t)$ // predict original image
11: $\quad \mathcal{L}_{\text{SRL}} \leftarrow -\mathcal{M}_{\text{RM}}(\hat{I}_0, I_{\text{ref}}^s)$ // calculate SRL loss with negative reward with Equation (3)
12: $\quad \mathcal{L} \leftarrow \mathcal{L}_{\text{Pre}} + \lambda \mathcal{L}_{\text{SRL}}$
13: $\quad \theta \leftarrow \theta - \eta \nabla_\theta \mathcal{L}$ // update model parameters via gradient descent ($\eta$ is learning rate)
14: **end for**

---

measured by either a VLM-based filter or the CSD model $\mathcal{M}_{\text{RM}}(\cdot)$ Somepalli et al. (2024); Xing et al. (2024). The reward loss is defined as:

$$\mathcal{L}_{\text{SRL}} = \mathbb{E}[\phi(\mathcal{M}_{\text{RM}}(I_{\text{ref}}^s, \hat{I}_0))] \tag{3}$$

where $\mathcal{Y} = \{y_i\}_{i=1}^n$ is the prompt set, $\phi$ maps reward scores to per-sample loss values, and $\hat{I}_0$ denotes the image generated by the diffusion model with parameters $\theta$ corresponding to prompt $y$.

To mitigate potential reward hacking, we jointly optimize the model by including the original Flow-Matching training objective, which is computed as:

$$\mathcal{L}_{\text{Pre}} = \mathbb{E}_{\boldsymbol{x}_0, t, \epsilon}[w(t)\|\boldsymbol{v}_\theta - \boldsymbol{v}_t\|^2] \tag{4}$$

where $w(t)$ is a weighting function, $\boldsymbol{v}_\theta$ denotes the neural network parameterized by $\theta$, and the sampling process is from $t = T$ with $\boldsymbol{x}_T \sim \mathcal{N}(\mathbf{0}, \boldsymbol{I})$ to $t = 0$, by solving the PF-ODE via $d\boldsymbol{x}_t = \boldsymbol{v}_\theta(\boldsymbol{x}_t, t)dt$. The final objective combines both losses:

$$\mathcal{L} = \mathcal{L}_{\text{Pre}} + \lambda \mathcal{L}_{\text{SRL}}, \quad \lambda = 0 \text{ before step } S, \ \lambda = 1 \text{ thereafter.} \tag{5}$$

As shown in Algorithm 1, we present the detailed SRL algorithm.

# 4 EXPERIMENTS

## 4.1 EXPERIMENTS SETTING

**USO Unified Benchmark.** To enable a comprehensive evaluation, we introduce USO-Bench, a unified benchmark built from 50 content images (20 human-centric, 30 object-centric) paired with 50 style references. We further craft 30 subject-driven prompts that span pose variation, descriptive stylization, and instructive stylization, along with 30 style-driven prompts. We generate four images per prompt for both subject-driven and style-driven tasks, and a single image for the combined style-subject-driven task. This yields 6000 samples for subject-driven generation, 7040 for style-driven generation, and 29500 for the combined task; full construction details are provided in the supplementary material.

**Evaluation Metrics.** For quantitative evaluation, we assess each task along three dimensions: **(1) subject consistency**, measured by the cosine similarity of CLIP-I and DINO embeddings following Wu et al. (2025c); **(2) style similarity**, reported via the CSD score Somepalli et al. (2024) for both style-driven and style-subject-driven generation, following Xing et al. (2024); and **(3) text–image alignment**, evaluated with CLIP-T across all three tasks.

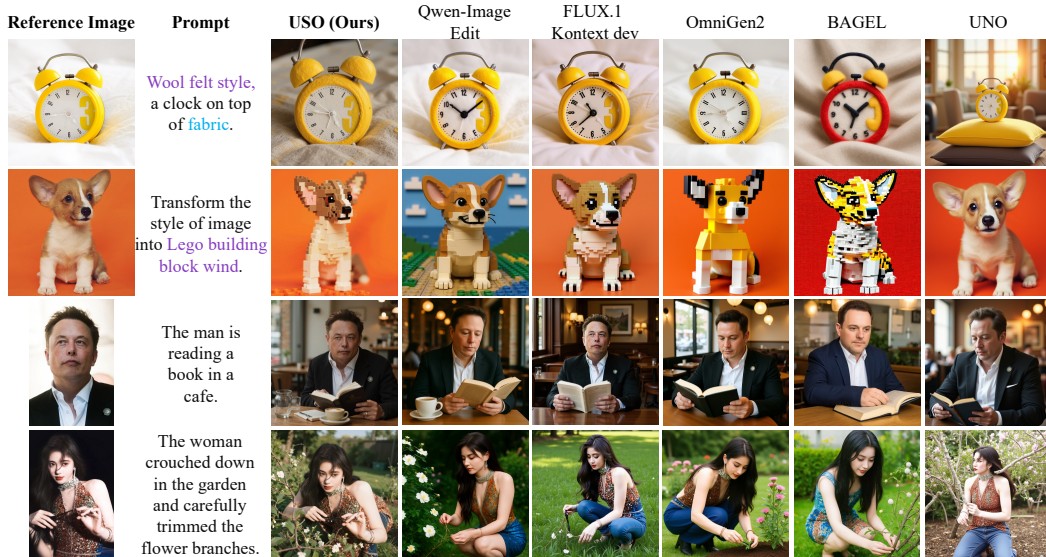

Figure 4: Qualitative comparison with different methods on subject-driven generation.

Table 1: Quantitative results for subject-driven and style-driven generation on USO-Bench.

| Method | Subject-driven generation | | | Style-driven generation | |
|---|---|---|---|---|---|
| | CLIP-I↑ | DINO↑ | CLIP-T↑ | CSD↑ | CLIP-T↑ |
| RealCustom++ Huang et al. (2024b) | 0.314 | 0.615 | **0.303** | - | - |
| RealGeneral Lin et al. (2025) | 0.485 | 0.732 | 0.275 | - | - |
| UNO Wu et al. (2025c) | 0.605 | 0.789 | 0.264 | - | - |
| BAGEL Deng et al. (2025) | 0.516 | 0.741 | 0.298 | - | - |
| OmniGen2 Wu et al. (2025b) | 0.475 | 0.723 | 0.302 | - | - |
| FLUX.1 Kontext dev Labs et al. (2025) | 0.579 | 0.775 | 0.287 | - | - |
| Qwen-Image Edit Wu et al. (2025a) | 0.544 | 0.756 | 0.302 | - | - |
| DEADiff Qi et al. (2024) | - | - | - | 0.462 | 0.274 |
| InstantStyle-XL Wang et al. (2024) | - | - | - | 0.540 | 0.276 |
| CSGO Xing et al. (2024) | - | - | - | 0.452 | 0.272 |
| StyleStudio Lei et al. (2025) | - | - | - | 0.348 | 0.282 |
| DreamO Mou et al. (2025) | 0.588 | 0.787 | 0.280 | 0.454 | 0.278 |
| **USO (Ours)** | **0.647** | **0.804** | 0.287 | **0.556** | **0.286** |

**Comparative Methods.** As a unified customization framework, USO is evaluated against both task-specific and unified baselines. For subject-driven generation, we benchmark RealCustom++ Mao et al. (2024), RealGeneral Lin et al. (2025), UNO Wu et al. (2025c), OmniGen2 Wu et al. (2025b), BAGEL Deng et al. (2025), FLUX.1 Kontext dev Labs et al. (2025), and Qwen-Image Edit Wu et al. (2025a). For style-driven generation, we compare StyleStudio Lei et al. (2025), DreamO Mou et al. (2025), CSGO Xing et al. (2024), InstantStyle Wang et al. (2024), and DEADiff Qi et al. (2024). For the joint style-subject-driven setting with dual conditioning, we compare OmniStyle Wang et al. (2025) and StyleID Chung et al. (2024).

## 4.2 EXPERIMENTAL RESULTS

**Subject-Driven Generation.** As shown in Figure 4, the first two rows demonstrate that USO simultaneously satisfies both descriptive and instructive style edits while maintaining high subject consistency. In contrast, competing methods either fail to apply the style or lose the subject. The last two rows further illustrate USO's strength in preserving human appearance and identity; it adheres strictly to the textual prompt and almost perfectly retains facial and bodily features, whereas other approaches fall short. When the prompt is "The man is reading a book in a cafe", FLUX.1 Kontext dev Labs et al. (2025) achieves decent facial similarity but carries copy-paste risks. As reported in Table 1, USO significantly outperforms prior work, achieving the highest DINO and CLIP-I scores and a leading CLIP-T score.

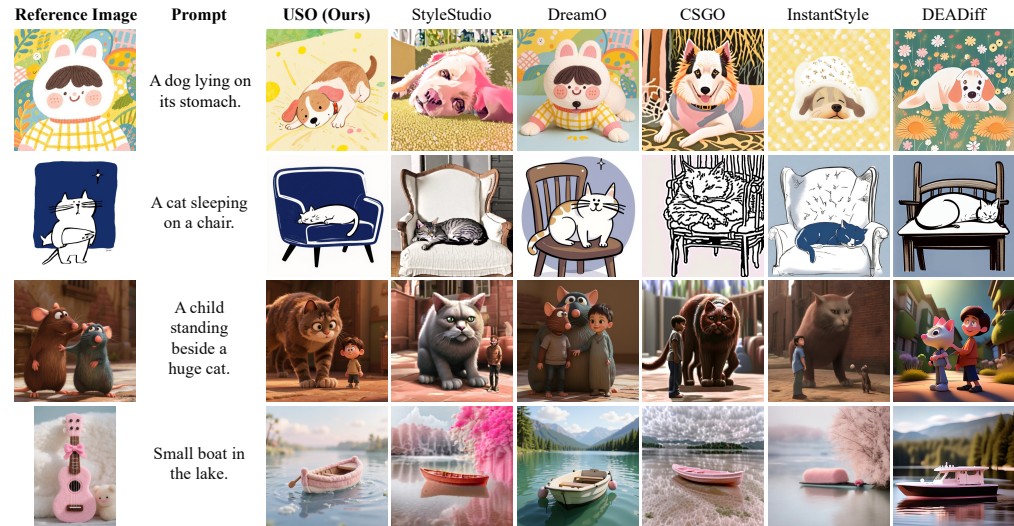

Figure 5: Qualitative comparison with different methods on style-driven generation.

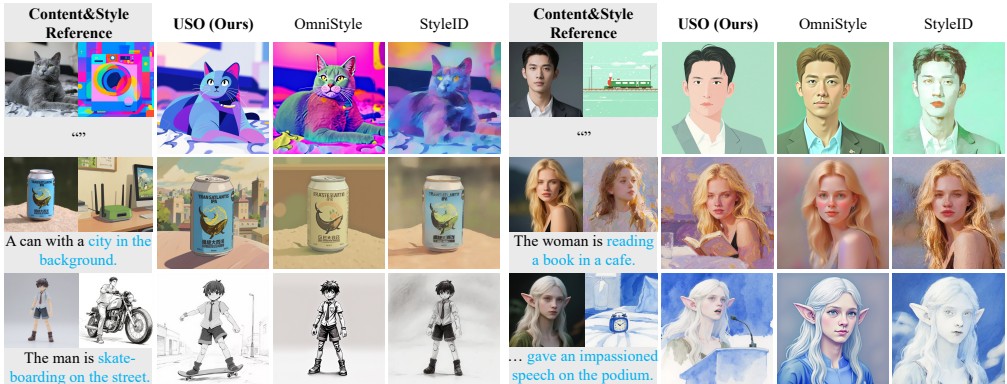

Figure 6: Qualitative comparison with different methods on style-subject-driven generation.

**Style-Driven Generation.** Figure 5 shows that USO outperforms task-specific baselines in preserving the original style, including global color palettes and painterly brushwork. In the last two rows, given highly abstract references such as material textures or Pixar-style renderings, USO handles them almost flawlessly while prior methods struggle, demonstrating the generalization power of our cross-task co-disentanglement. Quantitatively, Table 1 confirms that USO achieves the highest CSD and CLIP-T scores among all style-driven approaches.

**Style-Subject-Driven Generation.** As illustrated in Figure 6, we evaluate USO on both layout-preserved and layout-shifted scenarios. When the input prompt is empty, USO not only preserves the original layout of the content reference but also delivers the strongest style adherence. In the last two rows, under a more complex prompt, USO simultaneously

Table 2: Quantitative results for style-subject-driven generation on USO-Bench.

| Model | CSD↑ | CLIP-T↑ |
|---|---|---|
| StyleID Chung et al. (2024) | 0.407 | 0.230 |
| OmniStyle Wang et al. (2025) | 0.365 | 0.229 |
| **USO (Ours)** | **0.492** | **0.283** |

preserves the subject and identity consistency, matches the reference style, and aligns with the text, while other methods lag markedly and merely adhere to the text. Table 2 corroborates these observations, showing USO achieves the highest CSD and CLIP-T scores and substantially outperforms all baselines.

## 4.3 ABLATION STUDY

**Effect of style reward learning (SRL).** As shown in Figure 7(a), the middle column reveals a clear boost in style similarity for both style-driven and style-subject-driven tasks, with the identity

of the woman and the painting style closely matching the reference images. Removing SRL leads to a sharp drop in the CSD score and simultaneous declines in CLIP-I and CLIP-T, as reported in Table 3. We further visualize the reward curves in Figure 7(b) and Figure 7(c); our method yields improvements in both identity and style similarity. Notably, we rely **solely on style reward signals** and introduce **no identity-specific supervision**; nevertheless, the unified model gains in identity consistency. By sharpening the model's ability to extract and retain desired features, SRL brings overall improvements across all tasks, validating our motivation.

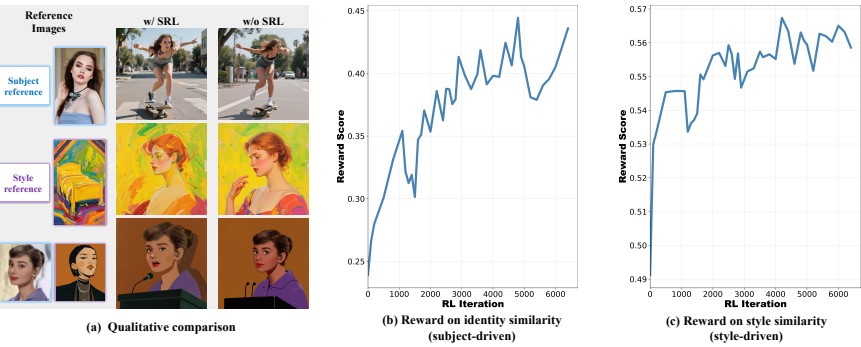

Figure 7: Ablation study of SRL.

Table 3: Ablation study of different components proposed in USO.

| Model | Subject-driven | | Style-driven | | Style-subject-driven | |
|---|---|---|---|---|---|---|
| | CLIP-I↑ | CLIP-T↑ | CSD↑ | CLIP-T↑ | CSD↑ | CLIP-T↑ |
| **USO (Ours)** | **0.647** | **0.287** | **0.556** | **0.286** | **0.492** | **0.283** |
| w/o SRL | 0.619 | 0.283 | 0.491 | 0.281 | 0.413 | 0.280 |
| w/o DE | 0.594 | 0.269 | 0.491 | 0.280 | 0.382 | 0.277 |

**Effect of disentangled encoder (DE).** Replacing the disentangled encoders with a single shared VAE to encode both style and content images degrades nearly every metric ( Table 3). We provide a qualitative comparison in Figure 10 of Section A.3.3.

**Effect of curated dataset.** As shown in Table 4, we reproduce two representative task-specific methods, UNO Wu et al. (2025c) and OmniStyle Wang et al. (2025), on our dataset to validate the effectiveness of the curated dataset. The reproduced OmniStyle even outperforms the original baseline, particularly in terms of CLIP-T, thanks to the layout-shifted triplets in the new dataset. Training UNO solely on the new dataset yields partial improvement, further confirming that both our method and the dataset contribute to the overall performance of USO.

Table 4: Quantitative results on USO-Bench. ∗ denotes models reproduced on our USO dataset.

| Model | Subject-driven | | Style-subject-driven | |
|---|---|---|---|---|
| | CLIP-I↑ | CLIP-T↑ | CSD↑ | CLIP-T↑ |
| **USO (Ours)** | **0.647** | **0.287** | **0.492** | **0.283** |
| UNO | 0.605 | 0.264 | - | - |
| UNO* | 0.596 | 0.278 | - | - |
| OmniStyle | - | - | 0.365 | 0.229 |
| OmniStyle* | - | - | 0.382 | 0.277 |

## 5 CONCLUSION

In this paper, we present USO, a unified framework capable of subject-driven, style-driven, and joint style-subject-driven generation. We introduce a cross-task co-disentanglement paradigm that first constructs a systematic triplet-curation pipeline, then applies content–style disentanglement training on the curated triplets to formulate a unified customization model. Additionally, we propose a style-reward learning paradigm to further boost performance. To comprehensively evaluate our method, we construct USO-Bench, a unified benchmark that provides both task-specific and joint evaluation for existing approaches. Finally, extensive experiments demonstrate that USO sets new state-of-the-art results on subject-driven, style-driven, and their joint style-subject-driven tasks, exhibiting superior subject consistency, style fidelity, and text controllability.

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

# A APPENDIX

## A.1 LLM USAGE STATEMENT

The core scientific ideas, methodology, experimental results, and conclusions presented in this paper are entirely the product of human authorship. A large language model was utilized solely as a language refinement tool, specifically to enhance the conciseness and clarity of the English text and to correct grammatical errors.

## A.2 EXPERIMENTS SETTING

### A.2.1 IMPLEMENTATION DETAILS.

We begin with FLUX.1 dev Labs (2024) and the SigLIP Zhai et al. (2023) pretrained model. We train on triplets $\{I_{ref}^c, I_{ref}^s, I_{tgt}\}$ for $21,000$ steps at batch size $64$, learning rate $8e - 5$, resolution $1024$ and reward steps $S = 18,000$. LoRA Hu et al. (2021) rank $128$ is used throughout.

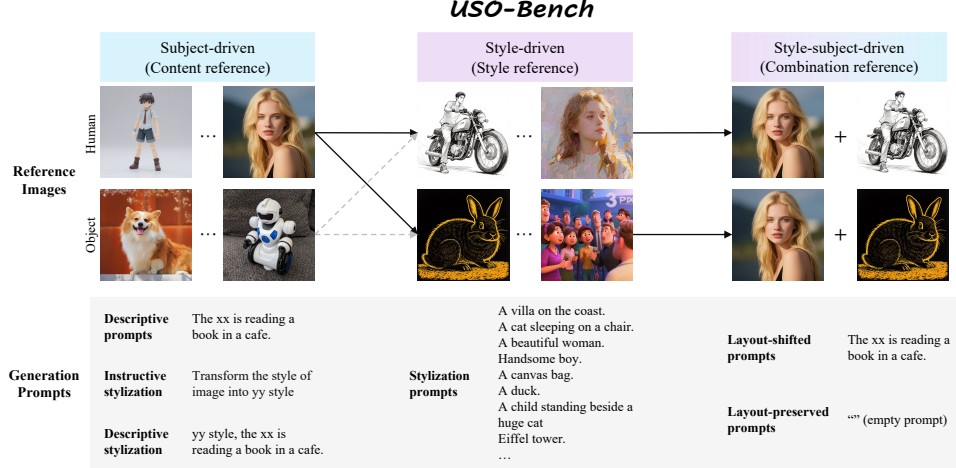

Figure 8: Examples of USO-Bench.

### A.2.2 DETAILS OF USO-BENCH.

USO-Bench is built to evaluate subject-driven, style-driven, and joint style-subject-driven generation. As shown in Figure 8, each subject-driven sample uses three prompt types: descriptive, instructive-stylization, and descriptive-stylization. By pairing these prompts with style-reference images from style-driven tasks, we obtain style-subject-driven samples via their Cartesian product. The resulting prompts are further split into layout-shifted and layout-preserved variants.

## A.3 ADDITIONAL EXPERIMENTS

### A.3.1 USER STUDY.

We further conduct an online user-study questionnaire to compare state-of-the-art subject-driven and style-driven methods. Questionnaires were distributed to both domain experts and non-experts, who ranked the best results for each task. (1) *Subject-driven tasks* were evaluated on text fidelity, visual appeal, subject consistency, and overall quality. (2) *Style-driven tasks* were judged on text fidelity, visual appeal, style similarity, and overall quality. As shown in Figure 9, our USO achieves top performance on both tasks, validating the effectiveness of our cross-task co-disentanglement and showcasing its capability to deliver state-of-the-art results.

| Method | DINO ↑ | CLIP-I ↑ | CLIP-T ↑ |
|---|---|---|---|
| Oracle(reference images) | 0.774 | 0.885 | - |
| Textual Inversion Gal et al. (2022) | 0.569 | 0.780 | 0.255 |
| DreamBooth Ruiz et al. (2023) | 0.668 | 0.803 | 0.305 |
| BLIP-Diffusion Li et al. (2023) | 0.670 | 0.805 | 0.302 |
| ELITE Wei et al. (2023) | 0.647 | 0.772 | 0.296 |
| Re-Imagen Chen et al. (2022) | 0.600 | 0.740 | 0.270 |
| BootPIGPurushwalkam et al. (2024) | 0.674 | 0.797 | 0.311 |
| SSR-EncoderZhang et al. (2024) | 0.612 | 0.821 | 0.308 |
| RealCustom++ Huang et al. (2024b); Mao et al. (2024) | 0.702 | 0.794 | **0.318** |
| OmniGen Xiao et al. (2024) | 0.693 | 0.801 | 0.315 |
| OminiControl Tan et al. (2024) | 0.684 | 0.799 | 0.312 |
| FLUX.1 IP-Adapter | 0.582 | 0.820 | 0.288 |
| UNO Wu et al. (2025c) | 0.760 | 0.835 | 0.304 |
| **USO (Ours)** | **0.800** | **0.838** | 0.316 |

Table 5: Quantitative results for single-subject driven generation on Dreambench Ruiz et al. (2023).

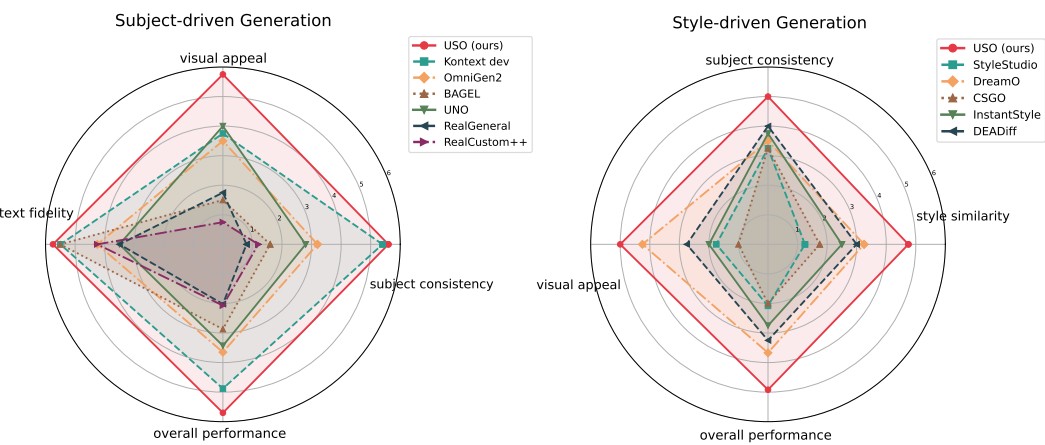

Figure 9: Radar charts of user evaluation of methods for subject-driven and style-driven generation on different dimensions.

### A.3.2 QUANTITATIVE EVALUATION ON DREAMBENCH RUIZ ET AL. (2023).

To further assess USO, we evaluate it on DreamBench Ruiz et al. (2023) in addition to USO-Bench. Following UNO Wu et al. (2025c), we generate six images per prompt, yielding 4,500 image groups across all subjects. As shown in Table 5, USO achieves the highest CLIP-I and DINO scores, and with a CLIP-T score of 0.316, it trails the top result (0.318) by only a narrow margin. These results demonstrate USO's superior subject consistency among state-of-the-art methods.

### A.3.3 ADDITIONAL ABLATION EXPERIMENTS

**Effect of Disentangled Encoder (DE).** We provide a visual comparison of using a single encoder versus separate encoders for the two conditions. As shown in Figure 10, the "cheetah" reverts to a photorealistic appearance, while the man's identity suffers a marked loss, further demonstrating the effectiveness of our disentanglement training.

**Effect of Hierarchical Projector.** To demonstrate the effectiveness of the Hierarchical Projector, we freeze all other parameters and fine-tune only this module to create a stylized variant that enables the pretrained T2I model to accept style-reference images as conditional input. This allows us to isolate its contribution. As shown in Table 6, the hierarchical projector achieves the highest CSD and a top CLIP-T score, confirming its key role in style-alignment training.

| Reference Images | w/ DE | w/o DE |

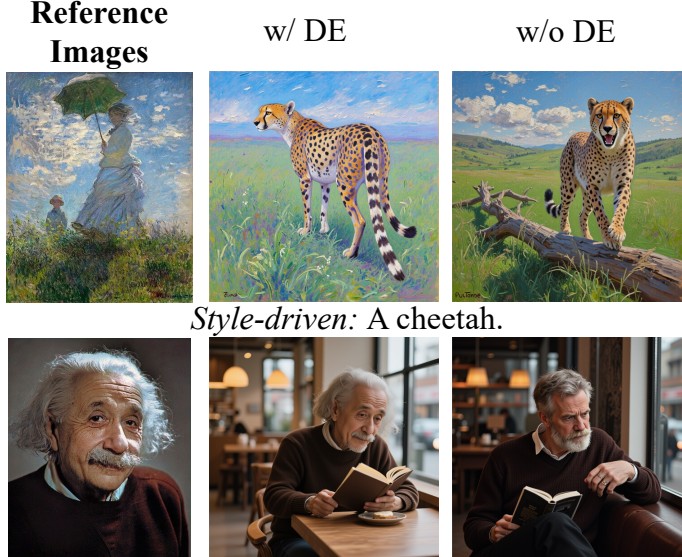

*Style-driven:* A cheetah.

*Subject-driven:* The man is reading a book in a cafe.

Figure 10: Ablation study of disentangled encoder. Zoom in for details.

Table 6: Ablation study of different projector in USO.

| Model | CSD↑ | CLIP-T↑ |
|---|---|---|
| resampler (depth=1) | 0.336 | 0.279 |
| resampler, unfreeze siglip | 0.155 | **0.288** |
| mlp (depth=1) | 0.277 | 0.284 |
| mlp, unfreeze siglip | 0.179 | **0.288** |
| hierarchical projector | **0.402** | 0.284 |

## A.4 MORE RESULTS.

We present additional qualitative results from USO:

- From Figures 11 to 14, USO demonstrates the ability to extract task-relevant content features while maintaining subject consistency across diverse textual prompts—capabilities that prior work typically treats as isolated tasks (e.g., subject-driven generation, instruction-based stylized editing, and identity preservation).

- In Figures 15 and 16, USO exhibits high stylistic fidelity, capturing both fine-grained characteristics (e.g., brushwork and material textures) and abstract artistic styles—far beyond simple color transfer.

- In Figures 17 and 18, USO freely combines arbitrary subjects with arbitrary styles, supporting both layout-preserving and layout-shifting generations.

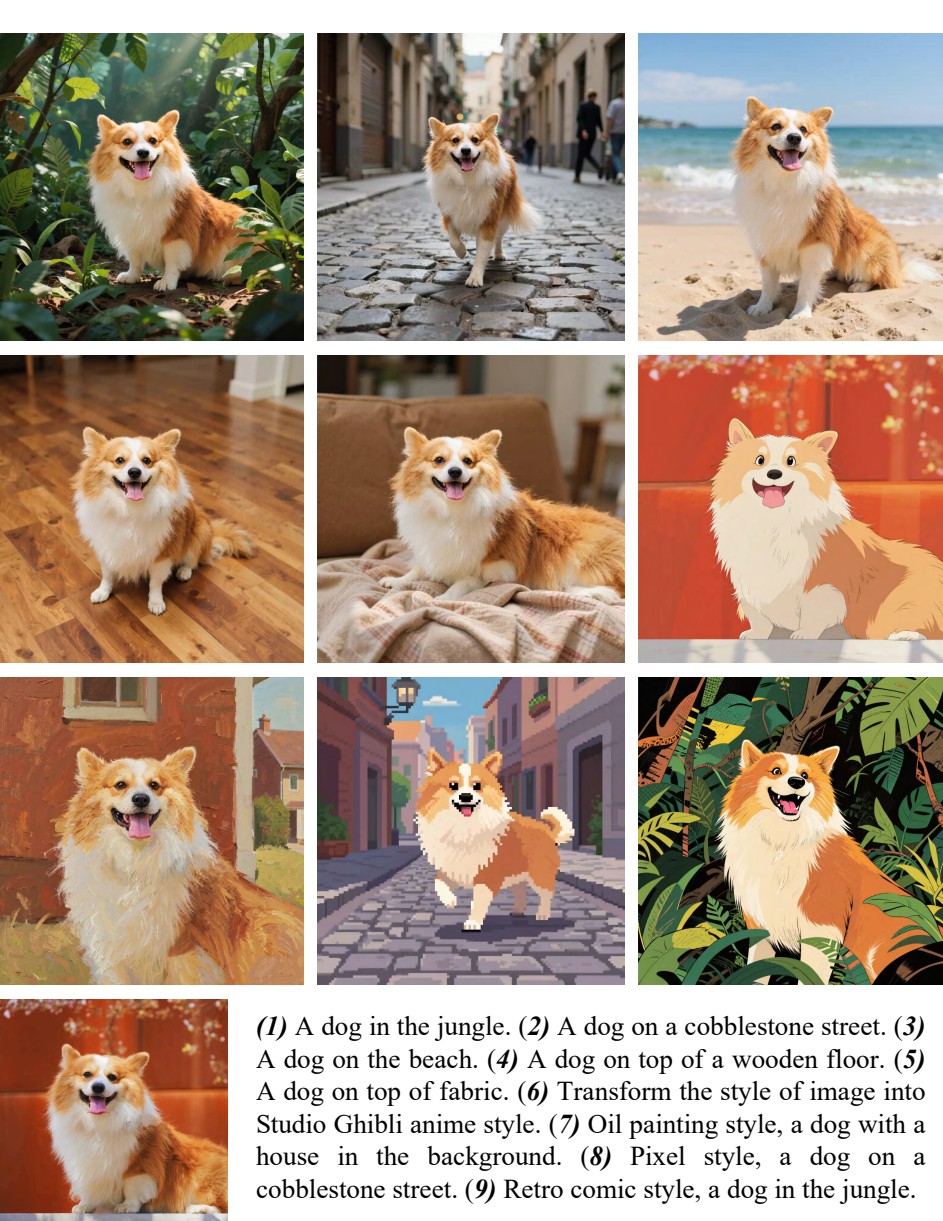

*(1)* A dog in the jungle. *(2)* A dog on a cobblestone street. *(3)* A dog on the beach. *(4)* A dog on top of a wooden floor. *(5)* A dog on top of fabric. *(6)* Transform the style of image into Studio Ghibli anime style. *(7)* Oil painting style, a dog with a house in the background. *(8)* Pixel style, a dog on a cobblestone street. *(9)* Retro comic style, a dog in the jungle.

Figure 11: More results on subject-driven generation.

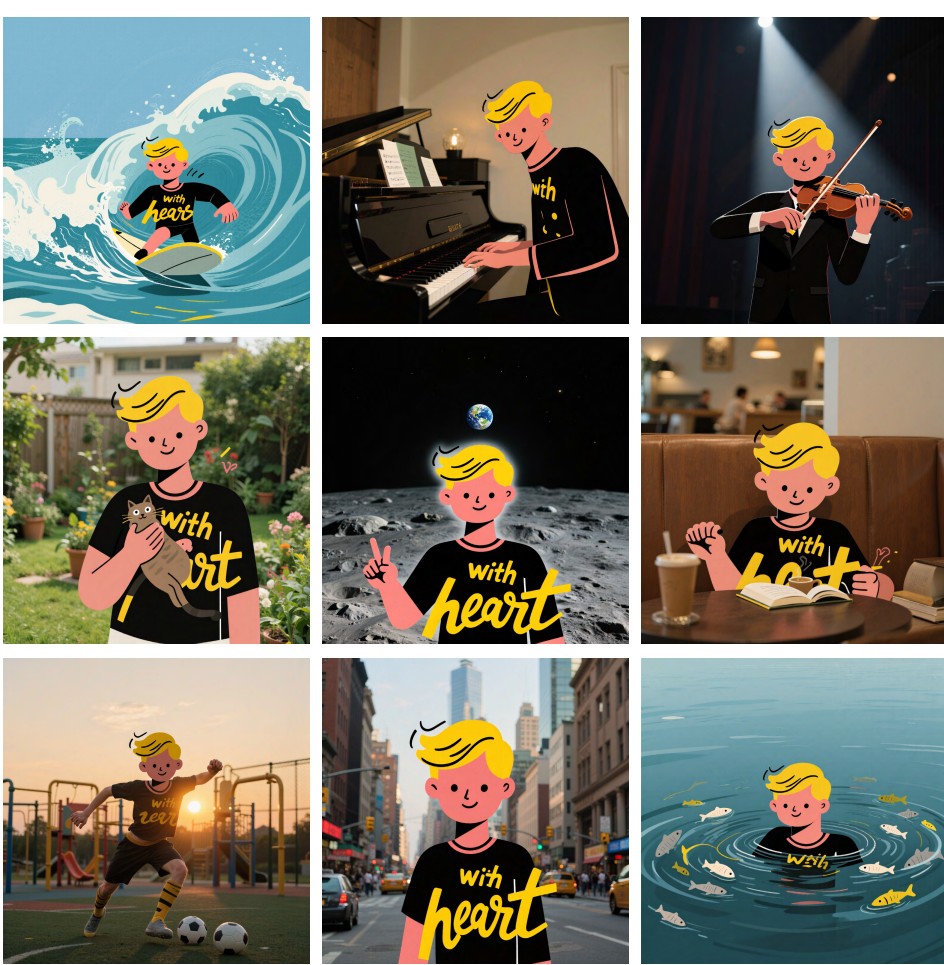

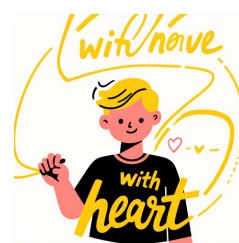 *(1)* This man is surfing, with the waves behind him chasing after him. *(2)* Handsome man is playing Piano. *(3)* This man in suit was playing the violin on the stage when a beam of light shone upon him. *(4)* This man is holding a cat in the garden. *(5)* This man stood on the moon and made a "yeah" sign, with a miniature of the Earth behind him. *(6)* The boy is reading a book in the coffe. *(7)* This man is playing football on the playground under the setting sun. *(8)* Handsome man in the city. *(9)* This man is in the water, with fish circling around him.

Figure 12: More results on subject-driven generation.

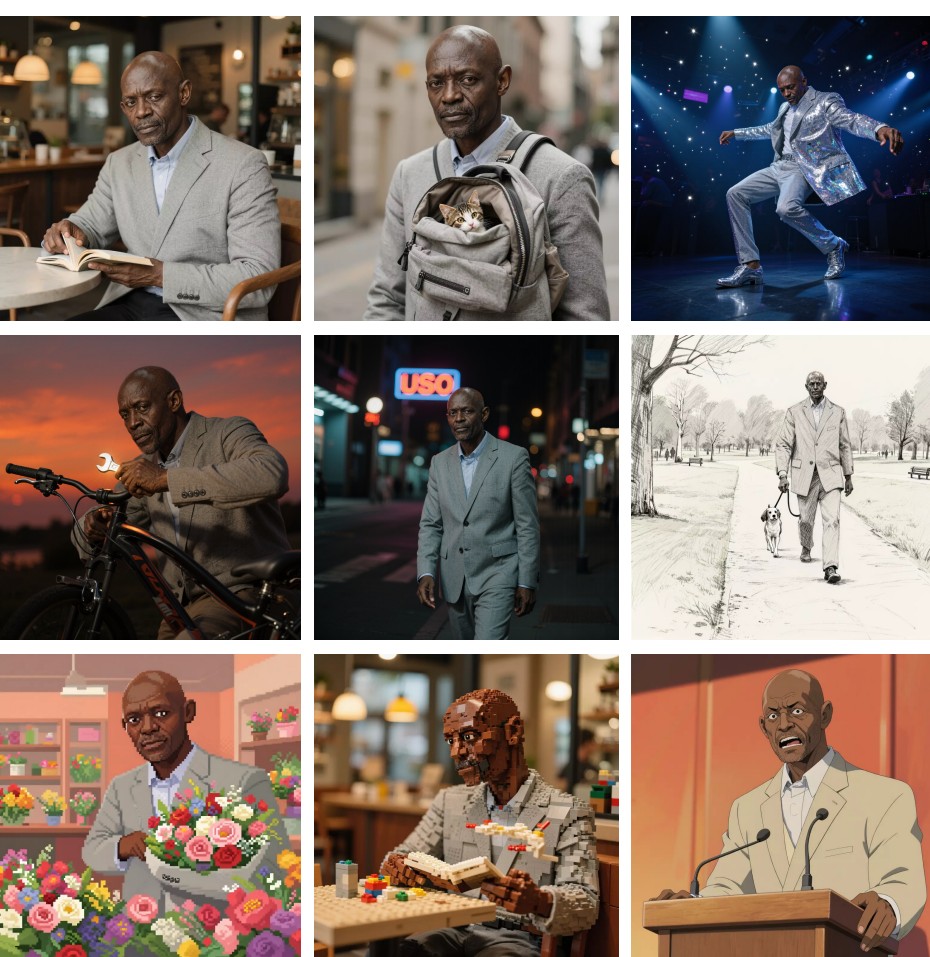

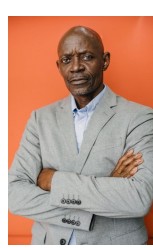

*(1)* The man is reading a book in a cafe. *(2)* The man carried a backpack with a kitten inside. *(3)* A man in a silver sequin jacket dances in a club, strobe lights bouncing off his coat like. *(4)* A man fixes a bike at dusk, wrench shining in orange twilight. *(5)* This man was walking on the street at night, with the blurry neon lights behind him reading "USO". *(6)* Sketch style, the man is walking with a dog, on the path in the park. *(7)* Pixel style, the man in flower shops carefully match bouquets, conveying beautiful emotions and blessings with flowers. *(8)* Lego building block wind, the man is reading a book in a cafe. *(9)* Studio Ghibli anime style, The man gave an impassioned speech on the podium.

Figure 13: More results on identity-driven generation.

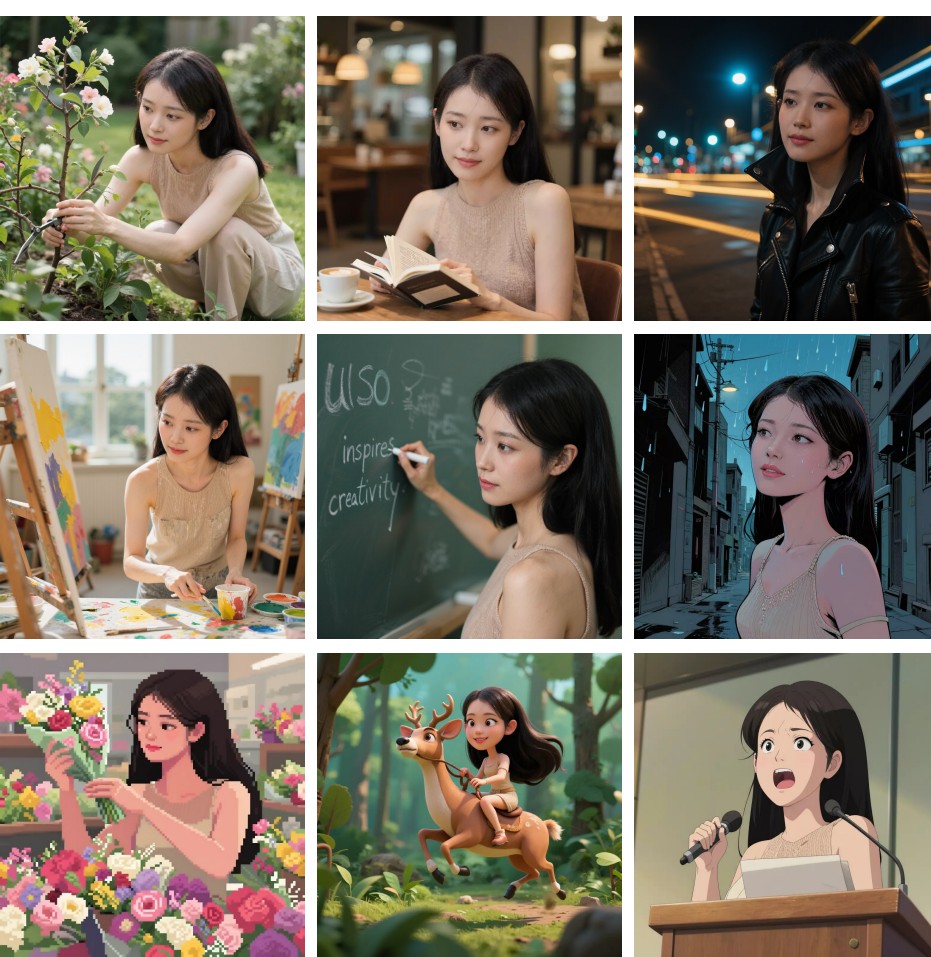

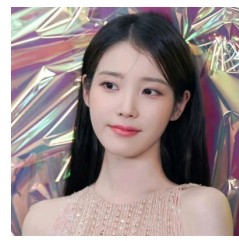

*(1)* The woman crouched down in the garden and carefully trimmed the flower branches. *(2)* The woman is reading a book in a cafe. *(3)* A woman in a black leather jacket at night, streetlights streaking past like gold lines, her jacket collar flipping to catch cool blue neon. *(4)* A woman is mixing paint in a sunny art studio. *(5)* This woman writes on the blackboard, side view, the blackboard blurs "USO inspires creativity". *(6)* Retro comic style, the woman is walking in a retro alley, with the sky drizzling and the raindrops clearly visible. *(7)* Pixel style, the woman crouched down in the garden and carefully trimmed the flower branches. *(8)* 3D Cartoon Style, the woman rides a deer in the forest. *(9)* Studio Ghibli anime style, the woman gave an impassioned speech on the podium.

Figure 14: More results on identity-driven generation.

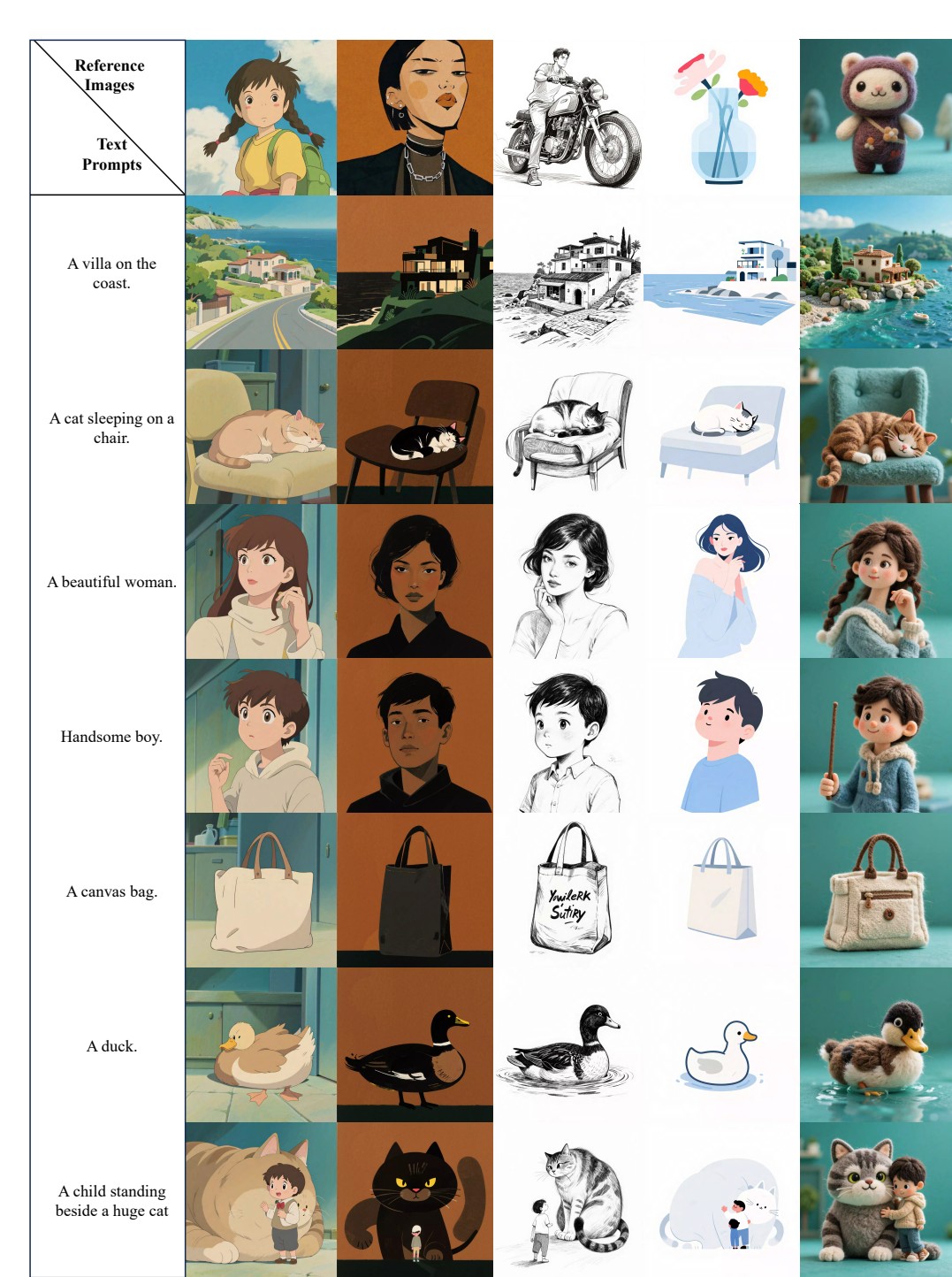

Figure 15: More results on style-driven generation.

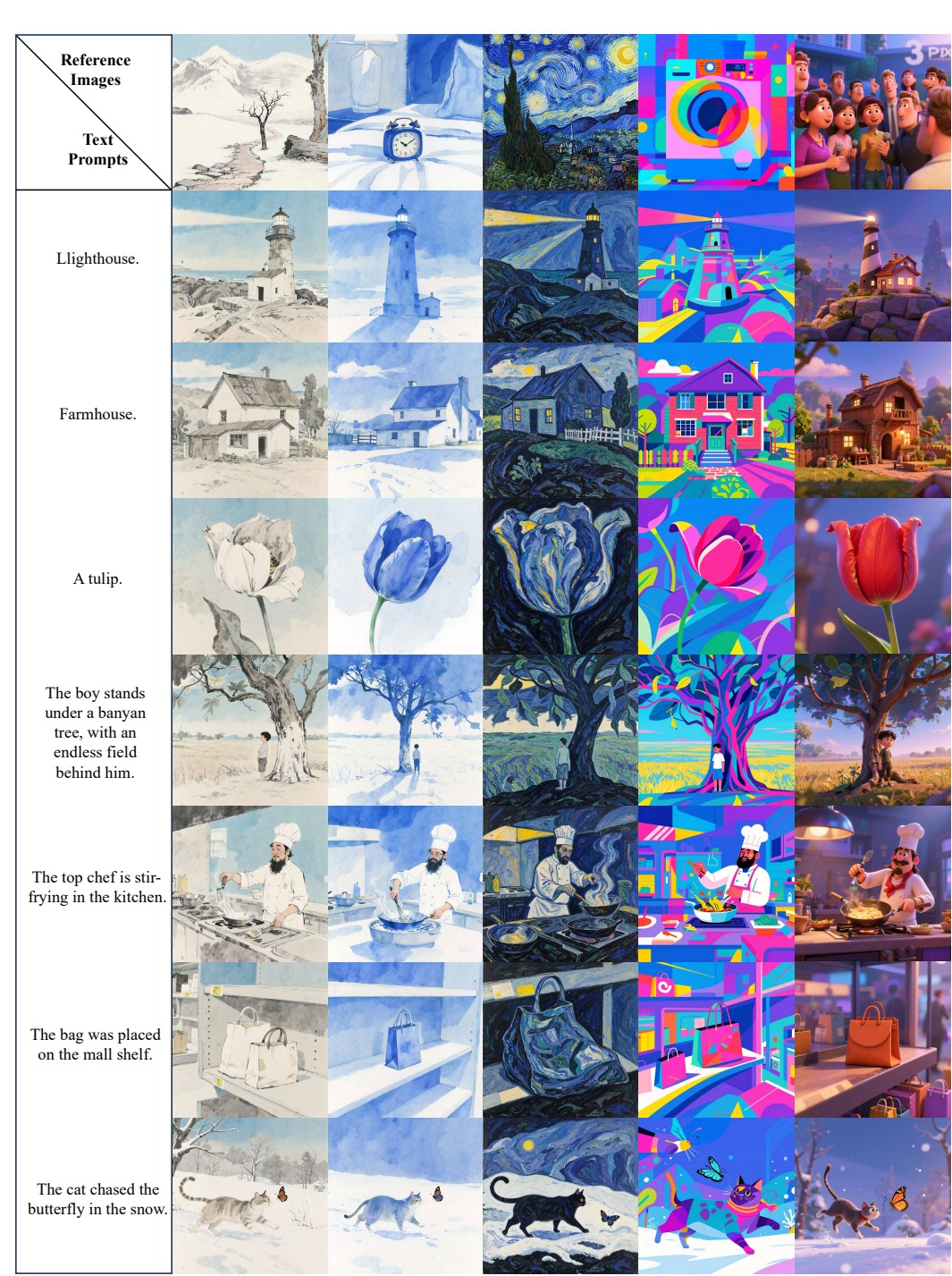

Figure 16: More results on style-driven generation.

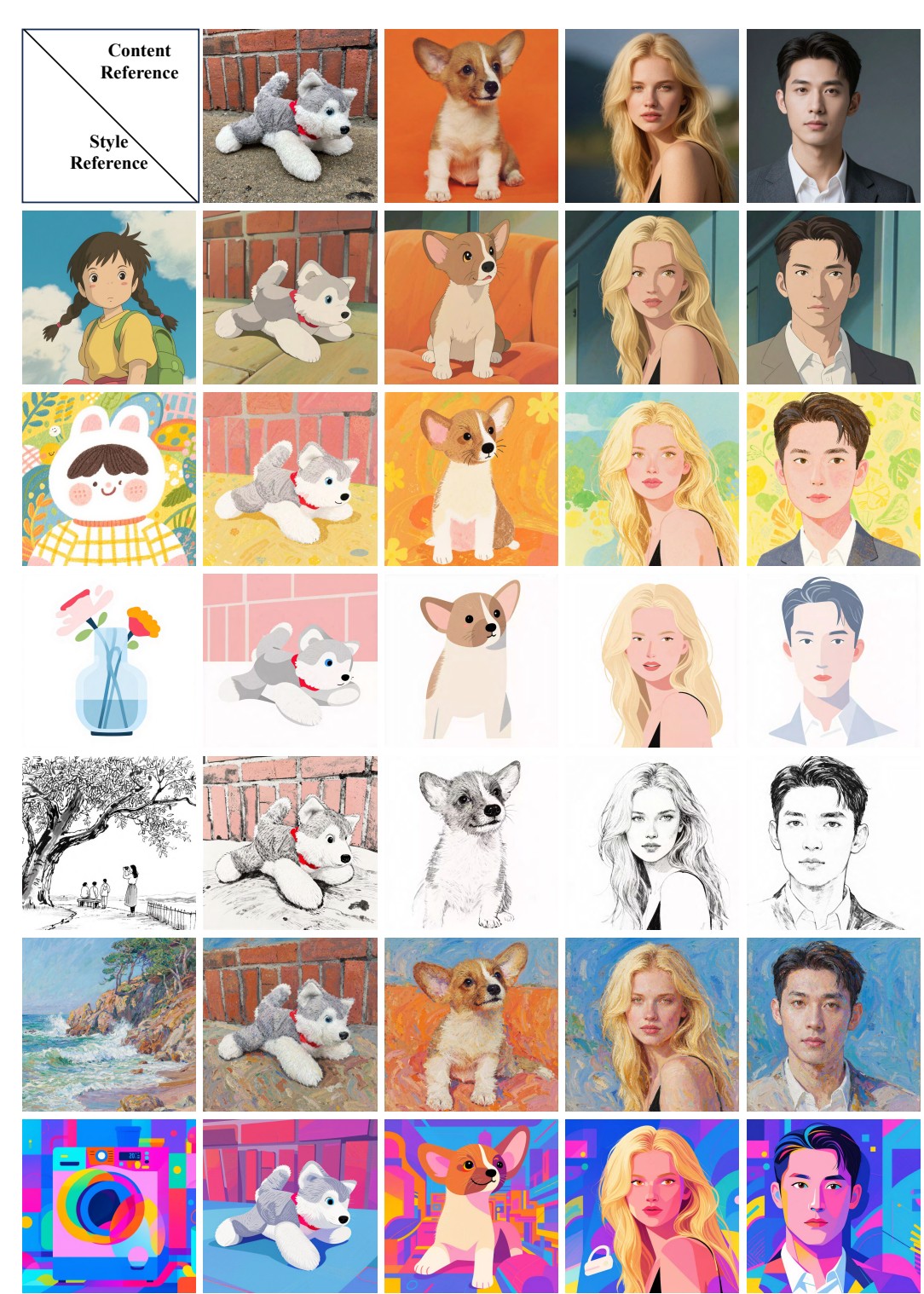

Figure 17: More results on style-subject-driven generation. We set prompt to empty for layout-preserved generation.

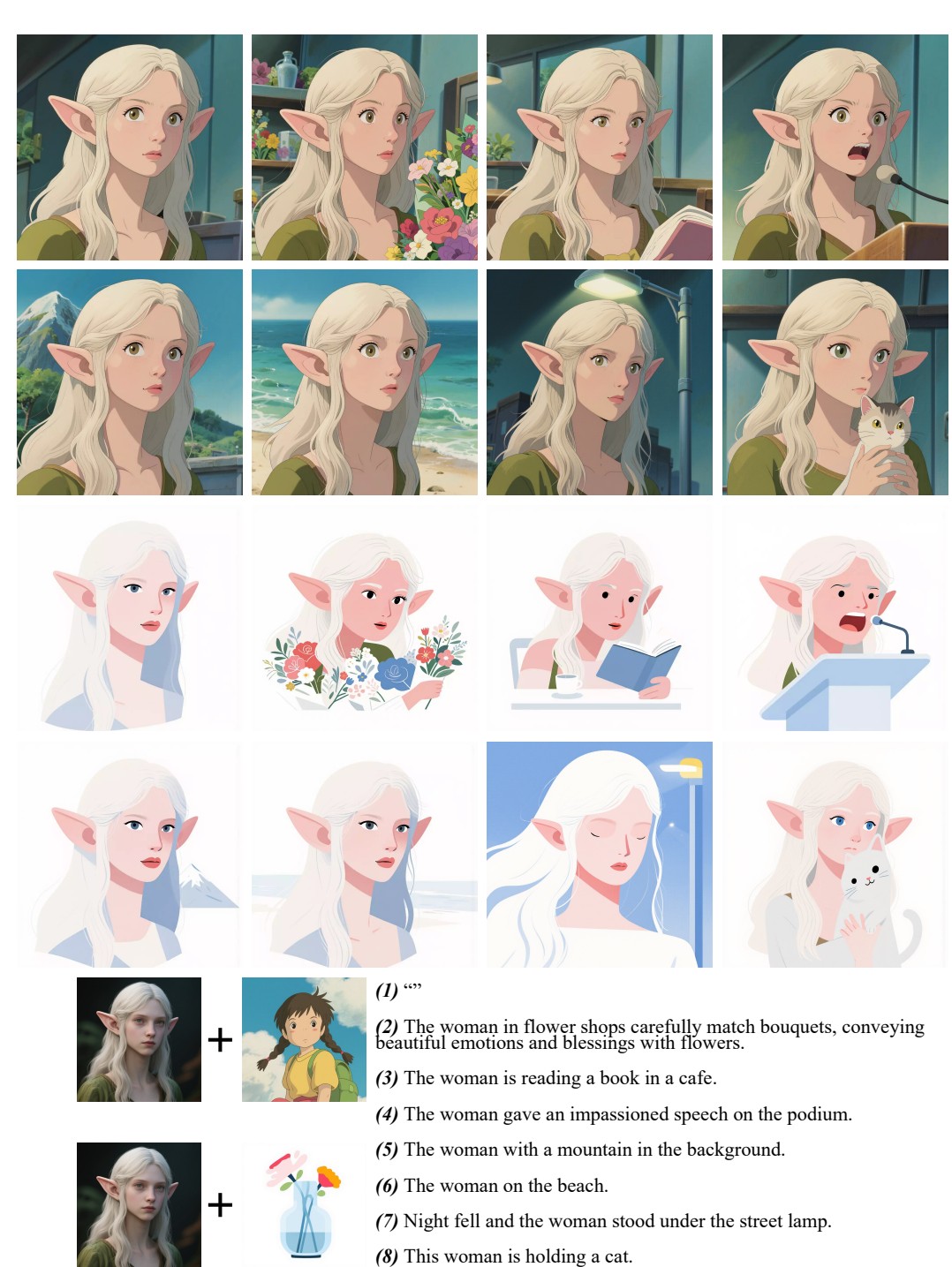

Figure 18: More results on style-subject-driven generation. USO supports any subject combined with any style in any scenario.