# OpenReview forum: "USO: Unified Style and Subject-Driven Generation via Disentangled and Reward Learning"
_ICLR.cc/2026/Conference — ICLR 2026 Conference Withdrawn Submission_

### Official Review · Reviewer_g3iS · 2025-10-15

**Soundness:** 2
**Presentation:** 2
**Contribution:** 1
**Rating:** 2
**Confidence:** 5

**Summary:**

USO addresses the isolation of style-driven generation  and subject-driven generation, which are treated as disjoint in previous research. The authors argue that these two objectives can be unified by leveraging disentanglement and recomposition of content and style and propose USO (Unified Style-subject Optimized customization model), a framework that enables mutual reinforcement between the two tasks rather than isolation.
The paper proposes

1. Cross-Task Triplet Curation Framework, which reverses the target style image to a style reference and a content reference.

2. Content-Style Disentangled Training, which is a quite standard in-context DiT, using SIGLIP to extract style feature, VAE to extract content feature.

3. Style Reward Learning,  which reinforces the model to prioritize desired style/content features by computing a reward (via VLM filters or the CSD model) between the generated stylized image and the style reference.

Experiments on USO-Bench and DreamBench show USO achieves state-of-the-art results in subject-driven generation, style-driven generation and style-subject-driven generation.

**Strengths:**

1. USO validated that  style-driven generation  and subject-driven generation could be jointly trained. The model obtain good results on subject-driven generation benchmarks.



2. The authors conducted rich experiments to demonstrate the effectiveness of the method.

3. The paper is organized well and overall easy to follow.

**Weaknesses:**

Major Weakness:
1. Cross-Task Triplet Curation Framework lacks novelty and misses many implementation details.

 First, the data synthesis framework lacks novelty. Presented in Figure 2, USO data curation framework reverses the style target image to generate style reference and content reference, which is not new because a similar idea of style removal module had been proposed and demonstrated effectively in 2023 [1].

 Second, many implementation details have been hided. For example, what is the performance of the style-expert model and de-stylization expert mdoel? With what data are they trained? In terms of style transfer ability itself, which one is better, the style expert model or USO? None of such information is revealed in the paper.

Third, it is a very strange setting to purposely shift the layout of content reference. Besides, even following this setting, the 'layout-shifted' results in Figure 1 and Figure 6 still demonstrate a strong copy-paste problem where the layouts are not obviously shifted.  Overall, this setting seems weird to me and even the purpose of the shifting layout is not achieved well by USO.

2. The USO training framework is  deficient in novelty. The overall structure is a standard in-context DiT training paradigm, with VAE encoder extracting content feature, SigLIP extracting style features. The authors claimed that style demands more semantic cues thus they employed the SigLIP encoder instead of VAE encoder, which seems lacking  persuasiveness to me. With what proof could not VAE  learn semantic style transfer? In fact, VAE definitely can and is well capable of learning semantic cues in style transfer, including 'geometric deformaton (3D cartoons)'. Thus the motivation of the USO design choice is  questionable. Besides, the hierarchical projector design lacks novelty, because it has been used by many  adapter-based image customization method, including styleshot[2], Ssr-encoder[3] in style transfer,  PuLID[4] in ID customization.


3. The proposed Style Reward Learning (SRL) is slightly adjusting existing reward-learning paradigm to style transfer scenario. SRL differs from ReFL  only in that ReFL focuses on "text fidelity or visual appeal" in text-to-image generation, while SRL is tailored for "reference-to-image setting". This is not a paradigm innovation but a scenario-specific adjustment. For example, both ReFL and SRL rely on external models (ReFL uses human preference models, SRL uses VLM/CSD models) to compute rewards, then backpropagate the reward signal to optimize the generator, which makes SRL a derivative work rather than an original design. Besides, the paper uses only existing and off-the-shelf similarity metrics (VLM-based filters or the CSD model) to compute this reward, with no novel design for the reward signal itself. This means the most critical part of SRL—the "reward definition"—lacks originality, as it depends entirely on pre-existing tools rather than introducing a new way to quantify "desired style features."  Finally, the reward learning may hurt the generalization ability of the in-context DiT model, making this model converges to a small number of biased style features.

Minor Weakness:

1. Some styles in the paper are not transfered well. For example, the stroke of the Van Gogh style is not well reconstructed, worse than many SDXL-based style transfer methods.

2. We could observe a significant color leakage problem in the style-subject-driven scenario, which consistently appears in all related figures in this paper. The color of the style reference image is always transfered to the background of the generated image, sometimes affecting the foreground subject as well.

3. The example triplet in Figure 3 shows that the reversely generated content reference still has a strong Chinese Ink style. The reverse content reference images in Figure 1 could not align with the target images very well. For example, there are more vegetables in the basket than the target image, the branch of bamboo is not aligned with target image.  Such miss-alignments may lead to inconsistent layout preservation.

Reference

[1] StyleDiffusion: Controllable Disentangled Style Transfer via Diffusion Models,  ICCV 2023

[2] StyleShot: A SnapShot on Any Style, arxiv

[3] SSR-Encoder: Encoding Selective Subject Representation for Subject-Driven Generatio, CVPR 2024

[4] PuLID: Pure and Lightning ID Customization via Contrastive Alignment, NeurIPS 2025

**Questions:**

1.  In terms of style-driven generation itself, which one is better, the style expert model or USO? The style expert is trained with how much data?

2. With what proof could not VAE  learn semantic style transfer well? This is generally not true.

3. Will the SRL cause USO to specialize on certain styles thus lose generalization capability?

4. What is the point of layout-shifted generation? This could be simply done by image customization/editing, then feed the result to layout-preserved style transfer. And in fact, the layout is not shifted obviously in the current version of USO.

---

### Official Review · Reviewer_hhUS · 2025-10-17

**Soundness:** 2
**Presentation:** 2
**Contribution:** 2
**Rating:** 4
**Confidence:** 5

**Summary:**

USO is a unified customization framework that jointly tackles style-driven and subject-driven image generation by disentangling and recombining content and style features. It builds cross-task triplets via stylization and de-stylization experts, then trains a DiT-based model with separate encoders plus stochastic conditioning dropout and Style Reward Learning to reinforce style extraction. On the new USO-Bench and DreamBench, USO attains state-of-the-art subject consistency, style fidelity, and text alignment.

**Strengths:**

1.USO is the first framework to unify subject-driven generation and style transfer.

2.USO achieves state-of-the-art subject fidelity and visual consistency on benchmark datasets.

**Weaknesses:**

1.CSD is trained on contrastive learning that uses artist name as the style label. There are discrepancy in the artworks for the single artist and most images in CSD's trained dataset are oil paintings, resulting un-reliable style reward score.

2.Proposed framework is similar to OmniStyle (replace VAE with SigLIP) and proposed reward learning is widely used in previous work, lack of novelty;

3.The paper shows that baselines fall shorts in text following in Figure 4, but USO do not achieve the best CLIP-T  score in Tabel 1.

4.The destylization model and dataset curation pipeline is similar to OmniStyle2[1].

5.Actually, SigLip is good at extract semantic feature with coupled style and content. The disentanglement in your paper is from dataset not the proposed encoder.

6.Lack of comparisons of SOTA style transfer methods like AttentionDistillation[2], AlignedGen[3].

[1]OmniStyle2: Scalable and High Quality Artistic Style Transfer Data Generation via Destylization.
[2]Attention Distillation: A Unified Approach to Visual Characteristics Transfer, CVPR 2025.
[3]AlignedGen: Aligning Style Across Generated Images, NIPS 2025.

**Questions:**

1.Could you provide more details about two experts models?

---

### Official Review · Reviewer_aT92 · 2025-10-27

**Soundness:** 2
**Presentation:** 2
**Contribution:** 3
**Rating:** 4
**Confidence:** 5

**Summary:**

This article mainly addresses the issue of style transfer while preserving both ID and style. It points out that previous methods could only maintain the layout and style of the subject, but failed to achieve adjustable ID. To solve this problem, the article proposes a data synthesis pipeline and common training methods.

**Strengths:**

1.The results look good.

2.The synthesis pipeline is relatively clear.

3.When compared with recent methods, the advancement of USO is evident.

4.It can handle various style control tasks.

**Weaknesses:**

The training details of the expert models in the two synthesis pipelines are not mentioned. Figure 2 is not clear.

The authors implemented USO based on UNO; to what extent does the achieved ID preservation capability stem from UNO?

When using CSD as the style reward, will it affect or even compromise the evaluation?

The authors did not measure the copy-paste phenomenon.

In some cases, the facial pose does not appear to change significantly. More reasonable supplementary cases may be needed.

The model parameters are not mentioned. What would the results be if the SDXL architecture is used? Most of the compared methods adopt the SDXL architecture.

**Questions:**

see Weaknesses

**Details Of Ethics Concerns:**

pictures of a lot of persons

---

### Official Review · Reviewer_G1wW · 2025-10-28

**Soundness:** 2
**Presentation:** 3
**Contribution:** 2
**Rating:** 6
**Confidence:** 4

**Summary:**

This paper proposed a Unified Style-Subject Optimized customization model, which treats style-driven and subject-driven generation as a unified task.

**Strengths:**

The core innovations of the described work include:
1. Unification of style and subject-driven generation under a single framework by addressing the disentanglement and recombination of content and style.
2. Construction of a large-scale triplet dataset with content images, style images, and stylized content images.
3. Implementation of a content-style disentangled learning scheme
4. Integration of a style reward-learning paradigm (SRL) to enhance model performance.

The expression in this paper is relatively clear.

**Weaknesses:**

1. Content Leakage in Style Encoder: The use of SigLIP as the style encoder is a notable weakness, as SigLIP features inherently encapsulate semantic information alongside style. This leads to potential content leakage, undermining the core objective of content-style disentanglement. A more dedicated style encoder should be considered to purify the style representation.

2. Lack of Explicit Content Consistency Constraint: The mechanism for preserving content consistency, particularly against layout shifts, is underspecified. Without an explicit constraint to align the target and content images, the model's reliance on the content input is unregulated. This results in unpredictable and potentially insufficient content fidelity in the outputs.

3. Uncontrolled Variable in Baseline Comparisons: The comparative analysis lacks discussion on the base models used by different methods. Since the performance of these generative models is heavily dependent on the capability of their underlying base models, the validity of the comparisons is compromised unless this variable is controlled for or thoroughly analyzed.

**Questions:**

1. Could you clarify whether the input text prompts contain explicit style descriptions? The understanding of what constitutes the "style" source is crucial for interpreting the results.

2. What specific measures are implemented in the model architecture or training objective to ensure content consistency, especially for cases involving significant layout changes between the content and target images?

3. Given the critical influence of base models, could you provide details on the specific base models used for each compared method in your experiments? Furthermore, have you conducted any ablation studies to isolate the contribution of your proposed algorithm from the choice of the base model?

---

### Note · Authors · 2025-11-14

I have read and agree with the venue's withdrawal policy on behalf of myself and my co-authors.